# Role of Equanimity on the Mediation Model of Neuroticism, Perceived Stress and Depressive Symptoms

**DOI:** 10.3390/healthcare9101300

**Published:** 2021-09-29

**Authors:** Nahathai Wongpakaran, Tinakon Wongpakaran, Danny Wedding, Zsuzsanna Mirnics, Zsuzsanna Kövi

**Affiliations:** 1Geriatric Psychiatry Unit and Psychotherapy Unit, Department of Psychiatry, Faculty of Medicine, Chiang Mia University, Chiang Mai 50200, Thailand; nahathai.wongpakaran@cmu.ac.th; 2School of Humanistic and Clinical Psychology, Saybrook University, Oakland, CA 94611, USA; danny.wedding@gmail.com; 3Institute of Psychology, Head of Department of Personality and Health Psychology, Károli Gáspár University of the Reformed Church in Hungary, Bécsi Street 324, H-1037 Budapest, Hungary; mirnics.zsuzsa@gmail.com

**Keywords:** depression, equanimity, strength, positive psychology, emotional instability, mindfulness, meditation

## Abstract

**Background:** Equanimity is widely and commonly practiced, but few have investigated the concept in clinical research. While the mediation model of neuroticism, perceived stress and depression have been demonstrated, it remains unclear whether equanimity mediates the relationship of these variables in parallel, serial or moderated mediation models. This study aimed to investigate the role of equanimity among those models. **Methods:** In all, 644 general participants (74.2% female, mean age = 28.28 (SD = 10.6)) provided data on the 10-item Perceived Stress Scale (PSS), the Neuroticism Inventory (NI), depression subscale of the Core Symptom Index, and the equanimity subscale of the inner Strength-based Inventory. Mediation and moderation analyses with the 5000 bootstrapping method were applied. **Results:** Equanimity was shown to moderate the relationship between NI/PSS and depressive symptom. Statistical evaluation supported all parallel, serial and moderated mediation models. Equanimity as a moderator provided a higher amount of percent variance explained by depressive symptoms than parallel and serial mediation models. **Conclusions:** Results suggest that the effect of perceived stress and neuroticism on depression can be mitigated by increasing levels of equanimity. The results demonstrated one potential benefit from practicing equanimity; enabling its extension to mental health problems could constitute an interesting focus for future research.

## 1. Introduction

Depression is a common psychological problem related to numerous risk factors. One of the risk factors associated with depression is the personality trait of neuroticism, one of the Big Five higher-order personality traits. Neuroticism is characterized by a tendency to experience negative emotions, such as anxiety, anger and fear [1]. The relationship between neuroticism and depression is well-established, with risk ratios and a 95% confidence interval of 1.25 (95% CI: 1.04, 1.45) [2]. The relationship between neuroticism and depression has been observed across cultures [3,4,5,6,7].

The effect of neuroticism on depression is complex, and multiple mediating variables have been investigated. These include catastrophic and anxiety-provoking appraisals [8], cognitive emotion regulation [9], social inhibition [6,10] and perceived stress [11]. Perceived stress, the feelings or thoughts that individuals experience following stressful life events, is independently linked and often precedes the occurrence of depression. Perceived stress has been shown to be a mediator of the effect of neuroticism on depression, and this effect is found across ages and in a broad range of populations [12,13,14,15,16,17,18,19,20,21,22]. Perceived stress is related to, and was believed to be, a key feature of neuroticism [23]. The relationship among neuroticism, perceived stress and depression has been documented in numerous studies [4,6,24,25,26,27].

The effect of neuroticism and perceived stress on depression may be reduced when positive variables are involved. For example, self-efficacy, “a belief in one’s capabilities to organize and execute the courses of action required to produce given attainments” [28], was found to significantly mediate the relationship between neuroticism and depressive symptoms [29]. Likewise, resilience was shown to have moderating and mediating roles in the associations of neuroticism and depressive symptom [30]. Mindfulness, a positive psychological attribute, has also been investigated to see how it affects neuroticism and depression. One study revealed that dispositional mindfulness partially mediated the relationship between neuroticism, posttraumatic stress disorders and depression symptoms [31,32]. Along with mindfulness, equanimity, a state of psychological stability and composure which is undisturbed by experience of or exposure to emotions, pain or other circumstances, usually co-occurs with mindfulness, and it is the authors’ primary interest. The virtue and value of equanimity are celebrated and recommended by a number of ancient philosophies and major religions, albeit different in concept and application. Buddhism emphasizes the virtue and cultivation of equanimity, and it is considered a refined state of mind based on freedom [33]. In addition to being mentioned in mindfulness or meditation, equanimity is a part of the main teaching and practice of the four divine abodes or the Immeasurable, i.e., loving-kindness, compassion, sympathetic joy or gladness and equanimity. It can also be found in the ten perfections and the seven enlightenment factors [34]. The ten perfections and the seven enlightenment factors are regularly practiced by devout Buddhists [33].

Clinically, equanimity may be perceived as a balanced emotional reaction toward stimuli, along with a tolerant and nonjudgmental attitude towards one’s fellow humans [35,36]. As equanimity is found to be integrated in mindfulness meditation method, it should hypothetically help control disruptive emotions among people with psychiatric problems. One study showed that equanimity results in fewer difficulties with emotional regulation, and this may explain the positive effect of mindfulness-based meditation on the emotional regulatory effect of mindfulness, as well as on neuroticism and alexithymia [32,37]. However, equanimity, although it is widely practiced in everyday life, is rarely taught or encouraged by healthcare providers [38], and studies focusing on equanimity and its clinical applications are still limited. One of the few studies to examine these relationships found that equanimity was associated with decreased anxiety and depression [39].

Equanimity is an integral part of emotional regulation, and it is believed to have an independent effect, increasing positive and decreasing negative psychological outcomes. Based on the aforementioned evidence, we hypothesized that equanimity may reduce levels of neuroticism, perceived stress and depression. To the best of our knowledge, no studies to date have investigated these relationships. In this study, we sought to explore the ways these variables interact. We also hypothesized that equanimity would act as a mediator or moderator in the relationship between neuroticism, perceived stress and depressive symptoms. We believe equanimity reduces the relationship between neuroticism and perceived stress on depressive symptoms, and we believe high levels of equanimity can reduce the effect of neuroticism and perceived stress on depressive symptoms.

## 2. Materials and Methods

### Study Population and Procedure

This study employed a cross-sectional design, including an online survey from December 2019 to September 2020 in Thailand. Convenience and snowball sampling strategies were applied to recruit the general Thai population via flyers, public websites and social media networks such as Facebook and LINE. A small payment was offered to each participant (100 THB or 3 USD). The inclusion criteria included (1) age ≥ 18 years, (2) able to understand, read and write Thai language, (3) able to use electronic questionnaires created by Google form, and (4) having their own electronic devices such as smartphone, tablet or notebook that can access the researchers’ online form. The exclusion criteria included (1) having a history of psychiatric disorders or being seen by a psychiatrist, and (2) being diagnosed or treated with substance use disorder. Sample size estimation based on correlation was calculated by determining Type I error (alpha) at 0.05, Type II error (beta, 1-power) at 0.20, with two-tailed test of significance. Expected effect size was set between small and medium (0.15), this yielded the minimum number of 343. However, based on the nature of online survey, the number of participants was expected to be higher, and the total sample was 649.

Each participant provided written informed consent before completing the questionnaire, which addressed sociodemographic data and related measurements. Ethics review and approval were obtained from the Faculty of Medicine, Chiang Mai University, Thailand.

## 3. Measurements

### 3.1. 10-Item Perceived Stress Scale (PSS-10)

This scale is used to measure to what extent the respondent has experienced stress during the last 4 weeks. It comprises a 10-item self-report using a 5-point Likert scale format (0 = never to 4 = very often), and the total score ranges from 0 to 40 [40]. Higher scores indicate greater perceived stress. The Thai version demonstrated good reliability and validity and has been widely used for both clinical and nonclinical sample [41]. In this study sample, the Cronbach’s alpha was 0.78.

### 3.2. Neuroticism Inventory (NI)

The NI is a dimensional measure of the neuroticism personality trait based on Eysenck’s five-factor model [4]. The NI, developed by Wongpakaran et al., [42] consists of a self-rating scale that includes 15 items with a 0 to 4 Likert scale. A higher score reflects a higher level of neuroticism. Cronbach’s alpha was 0.83. The NI was shown to have good validity and reliability. In this study sample, the Cronbach’s alpha was 0.90.

### 3.3. Core Symptom Index (CSI)

The CSI is used to measure common psychopathology in clinical practice. The CSI instructions directed respondents to answer the items based on how they felt over the past week. There are 15 CSI items: 5 items measure depression, 4 items assess anxiety and 6 items target somatization symptoms. Response options were based on a 5-point Likert scale, i.e., values of 0 (never), 1 (rarely), 2 (sometimes), 3 (frequently) and 4 (almost always). The higher the score, the higher the level of psychopathology. The CSI was shown to have good validity and reliability [43]. The depression subscale (CSI-D) was used in this study. In this study sample, the Cronbach’s alpha for CSI-D was 0.79.

### 3.4. Equanimity Scale (SBI-E)

The Equanimity scale is part of the 10 inner Strength-Based Inventory (SBI), e.g., truthfulness, generosity [44]. It is made up of one single item with 5 multiple-choice options, ranging from 1 to 5. The SBI item elicited optional outcome responses attributed to cognitive-emotional aspects of equanimity. For the equanimity item, the stem begins with “When I encounter losses/separations…” The choices include “It is very difficult for me to overcome losses, resulting in physical and mental symptoms, but not so serious that I need to go to the hospital.” Higher scores reflect a higher level of equanimity. The SBI-E correlated with other related strengths, e.g., positively related with patience and endurance (r = 0.164, *p* < 0.001). Because SBI-E is a single item, no internal consistency needs to be calculated. However, test–retest may be a better measure of participant consistency (or at least equally as informative as Cronbach’s alpha) [45]. The two-week test–retest reliability using intraclass correlation coefficient of the SBI-E was 0.88 (95% CI = 0.70, 0.95, *p* < 0.0001), indicating good reliability.

## 4. Statistical Analysis

Descriptive statistics—frequency, percentage, and mean and standard deviation—were obtained for sociodemographic characteristics. Mean and standard deviation were calculated for continuous data, e.g., NI, CSI-D, PSS and SBI-E scores. Correlation analysis was performed to determine the significant relationship between variables. Pearson’s correlation was used for the continuous variables, e.g., CSI-D and PSS; polychoric correlation was used for categorical or ordinal variables, e.g., sex and education, and polyserial correlation was used for categorical or ordinal and continuous variables, e.g., sex and neuroticism.

In checking data for mediation analysis, multiple regression was performed between variables. These analyses showed that the normal error distribution and homoscedasticity were met for all regressions indicating the validity of mediation analysis among these variables. To analyze the mediation and moderation models, we began by examining the magnitude of the relationships between neuroticism, depression, perceived stress and equanimity using zero-order correlations. For mediation analysis, we used the methods discussed by Hayes [46] to examine the relationship between neuroticism (X) and depression (Y) through perceived stress (M1). Furthermore, we tested the model when equanimity was included as the second mediator (M2) in a multiple parallel mediation model or serial mediation model (a causal chain linking the M1 and M2, with a specified direction of causal flow) (Figure 1).

For moderation analysis, we began by plotting the links between neuroticism (X) and depression (Y), between neuroticism (X) and perceived stress (M), and between perceived stress (M) and depression (Y), according to the high and low level of equanimity. Significant interaction was examined by visualizing predicted values of neuroticism or perceived stress scores with a high or low level of equanimity [46]. The moderation model that demonstrated the existence of a moderating effect would be included in the fully moderated moderation model. According to Hayes [46], if the moderation effect existed, then seven moderated mediation models could possibly be created; therefore, each model was tested (Figure A1).

Resampling or bootstrapping and the product of coefficients as suggested when conducting mediation and moderation analyses were performed [46,47]. We reported the results by standardized estimates, standard errors, *p*-values, confidence intervals for the direct effect coefficients and bootstrap confidence intervals for conditional indirect effects and for conditional indirect effects pairwise contrasts. Confidence intervals that did not straddle zero were indicative of statistical significance. For all the analyses, the level of significance was set at *p* < 0.05. All statistical analyses were performed using the program, IBM SPSS, 22.0. We used *M*plus 8.6 for all mediation and moderation analyses. MedCalc, Version 19.7 was used to create scatter plots and regression lines.

## 5. Results

Most participants were female; ages ranged from 18 to 72 (mean ± SD = 28.28 ± 10.6). Most lived alone (80.6%), earned a moderate income (62.9%) and had obtained a bachelor’s degree education level (70.8%). The clinical variable details are shown in Table 1.

Table 2 shows the correlation coefficients between variables CSI-D and SBI-E significantly correlated with age and marital status (*p* < 0.05). As expected, NI, CSI-D, PSS and SBI-E positively correlated with each other (*p* < 0.01). Only SBI-E had a negative correlation with all the remaining three variables.

Table 3 shows the estimate coefficients and standard error from each model starting from Model 1, the regression model without any mediator. Model 1 (Single regression model) shows that NI score predicted 32.6% of variance of the CSI-D variable. Models 2 and 3 (Mediation models) indicated the increased variances explained from 34.8% to 41.4% when the PSS score and SBI-E were added to the mediation model. NI had a significant indirect effect through PSS (β = 0.205, *p* < 0.001) and SBI-E (β = 0.043, *p* < 0.001), respectively. Models 4 and 5 denoted parallel and serial mediation models; NI had a significant indirect effect through both PSS and SBI-E (β = 0.008, *p* = 0.031) while both models explained the same 42.9% of variance for CSI-D. The alternative serial mediation model, where M1 was SBI-E and M2 was PSS, was also analyzed, and the results showed that the indirect effect of NI on CSI-D was similar to that of the original serial mediation model. 

Figure A2 shows that no significant difference was observed between slope of the regression line between PSS and NI. In the low level of equanimity, the slope coefficient was 0.291, whereas in the high level of equanimity, the slope coefficient was 0.352. No significant difference was noted between the two slopes (t = −0.88, *p* = 0.374).

Figure 2 shows a significant difference was observed between the slope of the regression line between PSS and CSI-D. In the low level of equanimity, the slope coefficient was 0.327, whereas in the high level of equanimity, the slope coefficient was 0.298. A significant difference was noted between two slopes (t = 3.048, *p* = 0.002).

Figure 3 shows a significant difference was observed between the slope of the regression line between NI and CSI-D. In the low level of equanimity, the slope coefficient was 0.341, whereas in the high level of equanimity, the slope coefficient was 0.254. A significant difference was noted between the two slopes (*t* = 2.243, *p* = 0.025).

After testing all seven moderation models, it appeared that Model 15, where SBI-E moderated the relationship between NI and CSI-D and between PSS and CSI-D, was best explained by conditional process. The figure illustrates the moderating effect of SBI-E in Model 15, indicating that the relationship between NI and CSI-D, and between PSS and CSI-D depended on the level of SBI-E (Figure 4).

Table 4 shows the difference of the indirect effects between two moderators. The model in which SBI-E moderated the relationship between PSS and CSI-D was found to have a larger effect size (3.47 for moderator 2 vs. 2.21 for moderator 1). The index of the moderated mediation model was significant (β = 0.384, *p* < 0.0001). The variance of CSI-D explained by this model was 64.6%.

## 6. Discussion

The present study investigated the extent to which neuroticism, perceived stress and equanimity influenced depressive symptoms. Both of our hypotheses regarding parallel/serial mediation and moderated mediation were confirmed in that high levels of equanimity were associated with decreased depressive symptoms. As shown by the increased R^2^ from 41.4% to 42.9% in parallel/serial mediation models, and 64.6% in the moderated mediation model, equanimity has a significant impact on depression.

As hypothesized, equanimity not only serves as a mediator for perceived stress or neuroticism and depression, but also as a moderator for those relationships. In other words, equanimity has a mediating effect on neuroticism and perceived stress, and has a buffering effect on depressive symptoms related to neuroticism and perceived stress. The fact that a variable like equanimity has both mediating and moderating effects has been found in other studies [48,49].

Equanimity is a character strength that can be learned or acquired, while neuroticism is a personality trait which is more likely to be stable and difficult to change. Our findings suggest that individuals characterized by high levels of neuroticism are prone to experience high perceived stress levels, but the development of depressive symptoms may be buffered by high equanimity levels.

The mechanisms involved in these relationships are unclear, and no prior study has been conducted on the effect of equanimity on these relationships. Evidence has shown that the practice of mindfulness and breathing meditation was negatively related to neuroticism [50,51]. Researchers also found that mindfulness meditation was directly related to equanimity [52,53], in line with the related investigation that the patients with personality problems who engaged in mindfulness practice may have cultivated a greater acceptance toward adversity and perceived less adversity and more equanimity in situations in which they felt vulnerable [54].

Our findings are in line with Juneau and colleagues in that neuroticism was negatively associated with equanimity [55]. However, we found that equanimity was associated with reduced depressive symptoms linked to neuroticism and perceived stress. Overall, both mindfulness and equanimity seemed to be closely linked. This finding has, in fact, already been demonstrated, especially when accessing deep levels of meditation [56,57]. What is added to the current knowledge is that equanimity appears to have direct effects on stress, neuroticism and depressive symptoms.

Some may wonder if equanimity can be cultivated among those who possess a personality trait for neuroticism. This could be conceptualized in terms of a learning process. Those with neuroticism tended to have worries and may have sought meditation training to calm themselves, which would be quite common in Thai society [58,59]. A study revealed that personality traits tend to be changed after mindfulness-based interventions, particularly among participants experiencing high levels of mindfulness [60]. Therefore, these participants with neuroticism may have had an opportunity to learn some psychological skills, including mindfulness and equanimity. The significant indirect effect of neuroticism through equanimity in the moderated mediation model indicated that those who acquired a high level of equanimity tended to have lower levels of depressive symptoms.

Since equanimity can be taught and learned by a variety of methods, future intervention studies of depression should focus on ways to increase equanimity, especially for those who have high levels of neuroticism and perceived stress. Equanimity is viewed as a significant virtue, and it has become one of the goals of psychological development. It can be learned through various kinds of meditation, e.g., mindfulness, breathing or by assimilating it as part of training in the four immeasurable attributes, e.g., loving-kindness, and other virtues such as patience, tolerance and wisdom (cognitive change in relation to one’s perceived experience, e.g., the awareness of reality of impermanence and being undisturbed by worldly vicissitudes) [33,34,61].

### Strengths and Limitations

To the best of our knowledge, this is one of the first studies to assess relationships among equanimity, neuroticism, perceived stress and depression. However, this study has limitations. It was conducted using a cross-sectional design; thus, we do not and cannot demonstrate cause–effect relationships. However, these cross-sectional results could set the stage for a longitudinal data analysis, and for addressing some related questions, such as the impact of other positive attributes on depression. The second point is that equanimity was measured using a single item, which may have resulted in differences from those studies that employed different equanimity measures. Finally, it would be impossible to completely eliminate the probability of a type I error in hypothesis testing. One common approach is to minimize the significance level of a hypothesis test to 1% (*p* < 0.01), which in this case, we found that the significant levels of the variables were all less than 0.01, suggesting the possibility of type I error was not too much to be concerned about. However, the possibility of a Type I error always exists, so replicating studies should be wary. If more studies show the same result, the evidence would be strongly confirmed.

## 7. Conclusions

Our results suggest that the effects of perceived stress and neuroticism on depression can be mitigated by enhancing equanimity. The results demonstrate the potential benefits from the practice of equanimity; this has implications for the treatment of mental illness, and it constitutes an interesting focus for future research.

## Figures and Tables

**Figure 1 healthcare-09-01300-f001:**
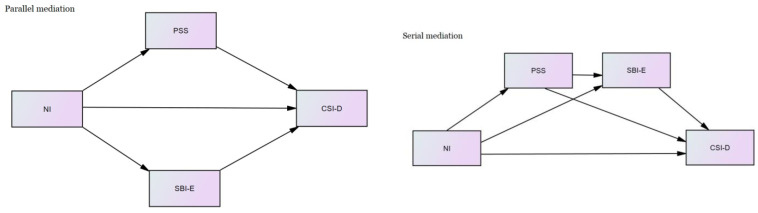
Proposed parallel and serial mediation models.

**Figure 2 healthcare-09-01300-f002:**
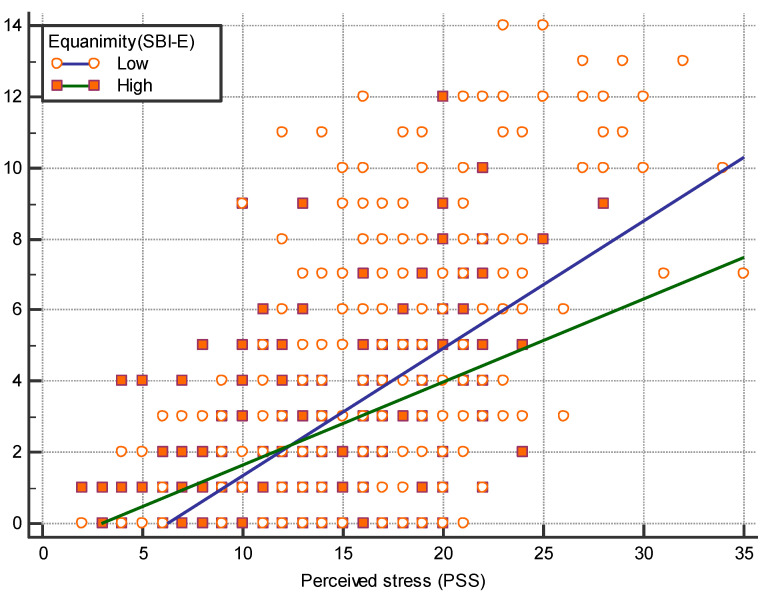
Regression lines between CSI-D and PSS scores based on level of equanimity.

**Figure 3 healthcare-09-01300-f003:**
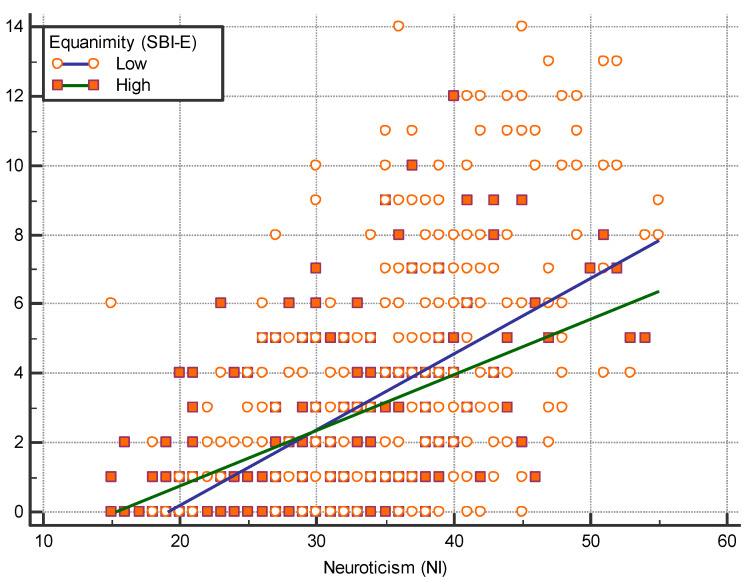
Regression lines between CSI-D and NI scores based on level of equanimity.

**Figure 4 healthcare-09-01300-f004:**
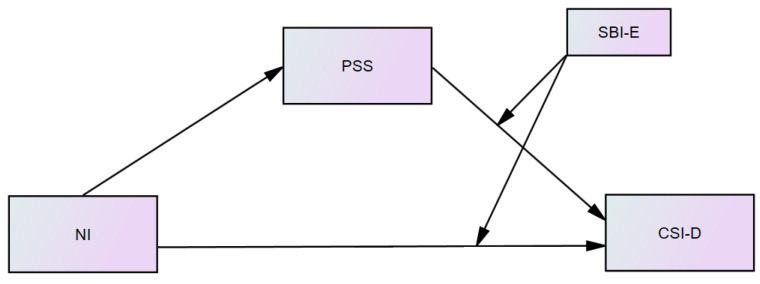
Final moderated mediation model (Model 15).

**Table 1 healthcare-09-01300-t001:** Socio-demographic characteristics of the participants (*n* = 644).

Variable		Value
Sex	*n* (%)	
	Female	478 (74.2)
	Male	166 (25.8)
Age	M ± SD	28.28 ± 10.6
Marital status	*n* (%)	
	Single/divorced/widowed	519 (80.6)
	Married	125 (19.4)
Monthly income	*n* (%)	
	Less than 20,000 THB	405 (62.9)
	20,000 and more	239 (37.1)
Education	*n* (%)	
	Less than Bachelor	67 (10.4)
	Bachelor	456 (70.8)
	Master and higher	121 (18.8)
Clinical variables	M ± SD	
	NI score	33.40 ± 9.0
	CSI-D score	3.05 ± 3.09
	PSS score	15.08 ± 6.0
	SBI-E score	3.81 ± 0.8

NI = Neuroticism Inventory, CSI-D = Depression scale of Core Symptom Index, PSS = Perceived stress scale, SBI-E = Equanimity scale of the inner Strength-based Inventory.

**Table 2 healthcare-09-01300-t002:** Correlation matrix between variables.

	1	2	3	4	5	6	7	8
1.Sex, female								
2.Age	0.009							
3.Marital, no partner	0.053	0.602 **						
4.Education, bachelor	0.197 *	0.267 **	0.288 **					
5.Monthly income, <700 US	−0.194 *	0.668 **	0.453 **	0.485 **				
6. NI score	0.030	0.002	0.012	−0.066	0.029			
7. CSI-D score	−0.026	−0.166 **	−0.168 **	−0.062	−0.090 *	0.578 **		
8. PSS score	0.004	−0.062	−0.001	−0.062	−0.054	0.599 **	0.611 **	
9. SBI-E score	0.217 **	0.128 **	0.098 *	0.110 **	0.053	−0.305 **	−0.355 **	−0.307 **

* *p* < 0.05 level ** *p* < 0.01, NI = Neuroticism Inventory, CSI-D = Depression scale of Core Symptom Index, PSS = Perceived stress scale, SBI-E = Equanimity scale of the inner Strength-based Inventory.

**Table 3 healthcare-09-01300-t003:** Mediation Effect of level of Perceived Stress and Equanimity on the Relationship Between level of Neuroticism and Depressive symptoms controlling for age, sex and education.

	Outcome: CSI-Dep	Product of Coefficients		*p*-Value	Bootstrapping Bias-Corrected 95%CI	
		Point Estimate	SE		Lower Limit	Upper Limit
Model 1 (R^2^ = 0.326) (Single regression model)	Total effect	0.571	0.024	0.000	0.531	0.608
	Direct effect					
	- NI	0.571	0.024	0.000	0.531	0.608
	Mediator: -					
	Total indirect	-	-	-	-	-
Model 2 (R^2^ = 0.414) (Mediation model)	Total effect	0.571	0.024	0.000	0.531	0.608
	Direct effect					
	Predictor-NI	0.366	0.033	0.000	0.313	0.422
	Mediator: PSS	0.361	0.036	0.000	0.301	0.418
	Total indirect	0.205	0.024	0.000	0.167	0.245
	Indirect effect					
	- via PSS	0.205	0.024	0.000	0.167	0.245
Model 3 (R^2^ = 0.348) (Mediation model)	Total effect	0.571	0.024	0.000	0.531	0.608
	Direct effect					
	- NI	0.528	0.026	0.000	0.484	0.568
	Mediator: SBI-E	−0.156	0.033	0.000	−0.210	−0.102
	Total indirect	0.043	0.011	0.000	0.028	0.063
	Indirect effect					
	- via SBI-E	0.043	0.026	0.000	0.028	0.063
Model 4 (R^2^ = 0.429) (Parallel mediation model)	Total effect	0.571	0.024	0.000	0.531	0.608
	Direct effect					
	- NI	0.340	0.033	0.000	0.287	0.397
	Mediator: PSS	0.346	0.036	0.000	0.285	0.403
	Mediator: SBI-E	−0.125	0.031	0.000	−0.177	−0.073
	Total indirect	0.231	0.024	0.000	0.192	0.273
	Indirect					
	- via PSS	0.196	0.024	0.000	0.159	0.237
	- via SBI-E	0.026	0.009	0.004	0.014	0.044
	- via PSS + SBI-E	0.008	0.004	0.031	0.003	0.017
Model 5 (R^2^ = 0.429) (Serial mediation model)	Total effect	0.571	0.024	0.000	0.528	0.604
	Direct effect					
	-NI	0.340	0.034	0.000	0.287	0.397
	Mediator: PSS	0.346	0.036	0.000	0.285	0.403
	Mediator: SBI-E	−0.125	0.031	0.000	−0.177	−0.073
	Total indirect	0.231	0.024	0.000	0.191	0.274
	Indirect					
	- via PSS	0.196	0.025	0.000	0.156	0.237
	- via SBI-E	0.026	0.009	0.004	0.014	0.044
	- via PSS + SBI-E	0.008	0.004	0.034	0.004	0.015

Note: NI = Neuroticism Inventory, CSI-D = Depression scale of Core Symptom Index, PSS = Perceived stress scale, SBI-E = Equanimity scale of the inner Strength-based Inventory.

**Table 4 healthcare-09-01300-t004:** Moderated Mediation Effect of Level of Perceived Stress and Equanimity on the Relationship between Level of Neuroticism and Depressive symptoms controlling for age, sex and education.

	Outcome: CSI-Dep	Product of Coefficients		*p*-Value	Bootstrapping Bias-Corrected 95%CI	
		Point Estimate	SE		Lower Limit	Upper Limit
Model 15 (R^2^ = 0.646)	Total effect	1.035	0.161	0.000	0.784	1.323
	Direct effect					
	- NI	0.651	0.059	0.001	0.651	1.009
	Mediator: PSS	0.676	0.095	0.000	0.474	0.785
	SBI-E	0.427	0.098	0.000	0.286	0.608
	Moderator1: SBI-E *Neuro	−0.418	0.189	0.027	−0.559	−0.178
	Moderator2: SBI-E *PSS	−0.413	0.119	0.001	−0.418	−0.130
	Total indirect	0.384	0.059	0.000	0.277	0.464
	Indirect effect					
	- via PSS	0.384	0.059	0.000	0.277	0.464

Note: NI = Neuroticism Inventory, CSI-D = Depression scale of Core Symptom Index, PSS = Perceived stress scale, SBI-E = Equanimity scale of the inner Strength-based Inventory.

## Data Availability

The datasets generated and/or analysed during the current study are not publicly available due to ethics ap-proval but are available from the corresponding author on reasonable request.

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
