# Peer review of "Role of Equanimity on the Mediation Model of Neuroticism, Perceived Stress and Depressive Symptoms"

_healthcare, 2021, doi:10.3390/healthcare9101300_

Round 1
Reviewer 1 Report
I am very interested in this kind of study, as it provides relevant information to guide clinical practice or to introduce specific elements in current interventions. However, I have some doubts and concerns regarding the quality of the manuscript in its current form. First, I think the authors have provided little information on the type of population that this study was targeting. In fact, it seems that it is a university population. Thus, I think they should provide additional information regarding how they targeted individuals. They have not provided information on whether they ruled out subjects or calculated the sample size they needed to carry out this study. Similarly, I would like to know if they discarded participants. If they do, they should clarify the reasons. It strikes me that the researchers did not receive a code from the ethics committee of their university. In the same way, it would be important to clarify whether this study adhere to Declaration of Helsinki. Given the exploratory nature of the study, it strikes me that they did not apply a statistical control procedure to avoid type 1 errors. I think they should delve into the limitations of the study, as well as the clinical relevance.
Author Response
Reviewer 1
- First, I think the authors have provided little information on the type of population that this study was targeting. In fact, it seems that it is a university population. Thus, I think they should provide additional information regarding how they targeted individuals.
Response The populations are general populations all over Thailand, not only university populations. We have added more information as follows. “The study was conducted in general Thai populations aged 18 years old”
- They have not provided information on whether they ruled out subjects or calculated the sample size they needed to carry out this study.
Response We have added more information regarding sample size calculation, and inclusion and exclusion criteria for the subjects as shown below.
This study employed a cross-sectional design, including an online survey from December 2019 to September 2020 in Thailand. Convenience and snowball sampling strategies were applied to recruit the general Thai population via flyers, public websites and social media networks such as Facebook and LINE. A small payment was offered to each participant (100 THB or 3 USD). The inclusion criteria included 1) age ≥ 18 years, 2) able to understand, read and write Thai language, 3) able to use electronic questionnaires created by Google form, and 4) having their own electronic devices such as smartphone, tablet, notebook that can access researcher’s online form. The exclusion criteria included 1) having a history of psychiatric disorders or being seen by a psychiatrist, and 2) diagnosed or treated with substance use disorder. Sample size estimation based on correlation was calculated by determining Type I error (alpha) at 0.05, Type II error (beta, 1-power) at 0.20, with two-tailed test of significance. Expected effect size was set between small and medium (0.15), this yielded the minimum number of 343. However, based on the nature of online survey, the number of participants was expected to be higher, and the total sample was 649.
Each participant provided written informed consent before completing the questionnaire, which addressed sociodemographic data and related measurements. Ethics review and approval were obtained from the Faculty of Medicine, Chiang Mai University, Thailand.
- Similarly, I would like to know if they discarded participants. If they do, they should clarify the reasons.
Response We have discarded some participants due to incomplete data. However, the number was small (5 participants). We have added the sentence as follows.
Of all 649, five were discarded due to incomplete data, therefore 644 was used for analysis.
- It strikes me that the researchers did not receive a code from the ethics committee of their university. In the same way, it would be important to clarify whether this study adhere to Declaration of Helsinki.
Response We did have an ethics code and adhered to the Declaration of Helsinki in the Institutional Review Board Statement section as shown below.
Institutional Review Board Statement: The study was conducted according to the guidelines of the Declaration of Helsinki and approved by the Institutional Review Board (or Ethics Committee) of the Faculty of Medicine, Chiang Mai University (study code, 184/2562 and date of approval, 8 July 2019).
- Given the exploratory nature of the study, it strikes me that they did not apply a statistical control procedure to avoid type 1 errors. I think they should delve into the limitations of the study, as well as the clinical relevance.
Response We thank the reviewer for pointing this out about Type I error. We have added this concern in limitation as detailed below.
Finally, it would be impossible to completely eliminate the probability of a type I error in hypothesis testing. One common approach is to minimize the significance level of a hypothesis test to 1% (p <0.01), which in this case, we found that the significant levels of the variables were all less than 0.01, suggesting the possibility of type I error was not too much to be concerned about. However, a possibility of a Type I error always exists, so replicating studies should be warrant. If more studies show the same result, the evidence would be strongly confirmed.

Reviewer 2 Report
Thank you very much for submitting your manuscript
“Equanimity on the Mediation Model of Neuroticism, 2 Perceived Stress and Depressive Symptoms”
for review and consideration for publication in
«Healthcare» (MDPI)
I sincerely appreciate the opportunity to review the manuscript.
Overall this is a rigorously designed online study and is likely to advance our understanding the role of perceived stress and neuroticism on depression. It “can be mitigated by increasing levels of quanimity”. So there is a potential benefit from practicing quanimity.
This manuscript is nicely drafted!
I have read it and had only one minor comment about it. Congrats for your work!
L36: Why do you use Buddhism when you don’t write anything in the whole abstract. Plus to me it is not obvious to think about this when reading about equanimity. Please consider to delete this. Why not using equianimity as a keyword?L67-76 you are referencing about Buddhism. That’s quite a small part.
I appreciate the opportunity to have read this manuscript and I thank you very much.
stay healthy and all the best
Author Response
Reviewer 2
Overall this is a rigorously designed online study and is likely to advance our understanding the role of perceived stress and neuroticism on depression. It “can be mitigated by increasing levels of quanimity”. So there is a potential benefit from practicing quanimity.
This manuscript is nicely drafted!
I have read it and had only one minor comment about it. Congrats for your work!
Response: We highly appreciate your feedback.
L36: Why do you use Buddhism when you don’t write anything in the whole abstract. Plus to me it is not obvious to think about this when reading about equanimity. Please consider to delete this. Why not using equianimity as a keyword?L67-76 you are referencing about Buddhism. That’s quite a small part.
Response: Yes, we agree with that. We deleted the word “Buddhism” from Keywords and replaced it with “equanimity” as suggested.
I appreciate the opportunity to have read this manuscript and I thank you very much.
stay healthy and all the best
Response: Thank you so much. We are so impressed and appreciate your attitude.

Round 2
Reviewer 1 Report
The authors have made an effort to answer the questions raised.